# The Impact of Overweight and Obesity on Reduced-Port Laparoscopic Distal Gastrectomy for Gastric Cancer Patients: A Propensity Score Matching Analysis of a Single-Institution Data

**DOI:** 10.3390/jcm11216453

**Published:** 2022-10-31

**Authors:** Ho-Goon Kim, Dong-Yeon Kang, Dong-Yi Kim

**Affiliations:** 1Department of Surgery, Division of Gastroenterologic Surgery, Chonnam National University Medical School, 42 Jebong-ro, Gwangju 61469, Korea; 2Department of Surgery, Division of Gastroenterologic Surgery, KS Hospital, 220 Wangbeodeul-ro, Gwangju 62248, Korea

**Keywords:** reduced port laparoscopic surgery, gastrectomy, overweight

## Abstract

This study aimed to investigate the short-term postoperative outcomes of reduced-port laparoscopic distal gastrectomy and demonstrate its safety and feasibility in overweight and obese patients with gastric cancer. The medical records of 211 patients who underwent reduced-port laparoscopic distal gastrectomy, between August 2014 and April 2020, were reviewed. After propensity score matching, they were divided into a non-overweight group (n = 68) and overweight group (n = 68). Operative details and short-term surgical outcomes were compared between two groups. Reduced-port laparoscopic distal gastrectomy in overweight group showed statistically longer operation time (200.59 vs. 208.68 min, *p* = 0.044), higher estimated bleeding volume (40.96 vs. 58.01 mL, *p* = 0.001), and lesser number of harvested lymph nodes (36.81 vs. 32.13, *p* = 0.039). However, no significant differences were found in hospital course and other surgical outcomes. There was no mortality in either group, and the postoperative morbidity rate was not significantly different (14.7% vs. 16.2%). In the subgroup analysis, overweight and obesity did not significantly affect postoperative complication rates (16.2% vs. 16.2%, *p* = 1). We demonstrated comparable short-term surgical outcomes of reduced-port laparoscopic distal gastrectomy between the two groups (*p* = 0.412~1). Reduced-port laparoscopic distal gastrectomy was safe in overweight and obese patients with gastric cancer.

## 1. Introduction

The advantages of laparoscopic gastrectomy over open gastrectomy have already been reported and are well accepted worldwide [1,2]. Today, laparoscopic surgery has become one of the natural treatment options for gastric cancer, and many surgeons now focus on minimizing its invasiveness. As part of that effort, there are increasing challenges of reduced-port or single port laparoscopic gastrectomy, of which feasibilities have been shown in several studies [3,4,5,6]. It was even demonstrated that reduced-port laparoscopic gastrectomy was oncologically safe and had comparable long-term outcomes to conventional methods [7,8]. Nevertheless, many surgeons are reluctant to perform laparoscopic surgery on overweight and obese patients. In terms of surgery for these patients with gastric cancer, several studies have reported that laparoscopic gastrectomy has no other inferiority except for a long operation time [9,10], but there are few studies on less invasive laparoscopic gastrectomy.

In our experience, patients with overweight and obesity had more difficulty in surgery than those who did not. An earlier study by Inagawa reported a significant difference in postoperative complications when a D2 resection was performed for gastrectomy with a BMI greater than 25 kg/m^2^ [11]. In addition, the World Health Organization defines obesity for East Asians as a BMI of ≥ 25 kg/m^2^. Thus, we compared two groups of patients who underwent gastrectomy: those with a body mass index ≥25 kg/m^2^ and those with a body mass index < 25 kg/m^2^.

We aimed to investigate the safety and feasibility of RpLDG using only three ports in overweight and obese patients (BMI ≥ 25 kg/m^2^) with gastric cancer. Propensity score matching (PSM) analysis was introduced to reduce selection bias in observational studies, and treatment effects are considered to be roughly randomized, mimicking randomized controlled trials [12]. Therefore, we endeavored to reinforce the validity of the results by implementing PSM analysis.

## 2. Materials and Methods

### 2.1. Patients

In our institution, RpLDG using three ports had been offered to gastric cancer patients with early gastric cancer and limited advanced gastric cancer since 2014. The indication of RpLDG was clinical stage cT1-3N0. If lymph node metastasis was suspected but the clinical stage was T2N1 or T3N1, it was included in the indication according to the preference of a patient. Overweight and obesity were not considered an exclusion criterion. Between August 2014 and April 2020, 270 patients underwent RpLDG for gastric malignancies. Of these, excluding 17 patients with combined surgery for other organs and 42 patients with additional trocars, 211 patients were retrospectively enrolled in the present study. All RpLDGs were performed by a single surgeon with experience of ≥60 gastrectomies/year. Before initial data collection for this study, the surgeon already had experience of a total of more than 350 cases of laparoscopic gastric cancer surgery. The gastric malignancy was diagnosed by esophagogastroduodenoscopy with biopsy. Preoperative clinical stage was evaluated through computed tomography (CT), routinely performed in all patients. Positron emission tomography-CT or magnetic resonance imaging was performed in selected patients for clinical staging. We divided the patients into two groups based on a BMI of 25 kg/m^2^; 143 patients with a BMI < 25 kg/m^2^ were classified into the non-overweight group (NOG) and 68 patients with a BMI ≥ 25 kg/m^2^ were classified into the overweight group (OG). All patients were treated postoperatively according to a uniform clinical pathway. The key steps of the clinical pathway were no use of nasogastric tube, early mobilization, and resumption of water feeding on postoperatively day (POD) 1 and enteral feeding from POD 2. Discharge was routinely recommended on POD 6 for patients with the following discharge criteria: no abnormal physical signs, laboratory test results within acceptable levels, adequate pain control without intravenous or oral analgesics, and tolerance of oral intake without adverse gastrointestinal symptoms. However, in some cases, discharge was delayed because of patients’ own personal reasons.

### 2.2. Data Collection and Definition

This retrospective study was approved by the Institutional Review Board of Chonnam National University Hospital, Gwangju, Korea, who waived the requirement for informed consent (IRB No. CNUH-2021-353). The hospital’s prospective data registry was reviewed to collect clinicopathological characteristics, operative details, and record of hospital course. After collecting data, PSM analysis was then performed. RpLDG was defined as the operation performed by an operator and a laparoscopist using only three abdominal ports, which was introduced in our previous study [4]. Surgery was initiated with three ports, but the case of adding a port was defined as a conversion and was considered an exclusion criterion. Postoperative morbidity and mortality were defined as any events occurring within 30 days of the operation or during hospitalization. A range of postoperative complications (including cardiovascular, pulmonary, and renal problems), systemic complications (including ascites, chyle leakage, gastric stasis, ileus, intraabdominal infection, pancreatitis and wound infection), and surgical complications were compared between the two groups. The severity of the complications was graded using the Clavien-Dindo classification [13]. The pathological tumor staging was based on the seventh edition of the Union Internationale Contre le Cancer/American Joint Committee on Cancer TNM classification system [14].

### 2.3. Surgical Technique

The patients were placed in the reverse Trendelenburg position, with their legs apart. Two operator working ports were applied on the right subcostal area (5 mm port) and right mid-abdomen (12 mm port). The laparoscopist stood between the legs of the patients and used an umbilical port for the laparoscope and pneumoperitoneum. The liver was retracted upward using a straight needle with a nylon string, which passed percutaneously and was fixed on the gastrohepatic ligament with a polymer locking clip. The regional lymph node dissection was performed as described in the Japanese Gastric Cancer Treatment Guidelines [15] using the LigaSure™ (Valleylab, Boulder, CO, USA) or the Harmonic Ace 1 (Ethicon Endo-Surgery, Cincinnati, OH, USA). D1+ lymph node dissection was generally performed, with D2 dissection selected for cases of ≥cT2 tumors or suspected lymph node metastasis. During dissection of the suprapancreatic area, the pancreas was compressed with the right hand. A gauze was used to prevent the stomach from moving downward for dissection of lymph node No. 11p. In early RpLDG, we performed extracorporeal anastomosis which was similar to that performed in conventional LDG. After adoption of intracorporeal anastomosis, most patients in the later period of RpLDG underwent intracorporeal Billroth-II or Roux-en-Y anastomosis to eliminate the need for mini-laparotomy anastomosis. For extracorporeal anastomosis, a 5 to 6 cm epigastric incision was made, and distal gastrectomy and Billroth-I, -II, or Roux-en-Y gastrojejunostomy was performed through the incision. For intracorporeal anastomosis, reconstruction excluding Billroth-I was performed using a Powered endopath stapler (Ethicon, San Francisco, CA, USA) or Signia™ stapler (Medtronic, Mansfield, MA, USA), and the umbilical port was extended to 2 to 3 cm to extract the specimen.

### 2.4. PSM

To minimize selection bias and balance any differences in baseline characteristics, PSM was conducted between the two groups. The baseline information was matched including age, sex, abdominal surgical history, comorbidity, American Society of Anesthesiologists (ASA) score and pathological features of the tumor. The nearest neighbor matching was performed in a 1:1 ratio without replacement and a caliper width of 0.01 standard deviation was specified.

### 2.5. Statistical Analysis

Continuous variables and numerical data were presented as mean ± standard deviation, and compared using Student’s *t*-test or Mann–Whitney test after verifying the normality of distribution using a Kolmogorov–Smirnov test. Categorical variables were described as numbers (%) using Pearson’s chi-square test or two-tailed Fisher’s exact test. PSM was performed using the variables in Table 1 except for BMI as a correction variable. Hosmer-Lemeshow test verified that the model was well calibrated (*p* = 0.251). We presented the *p*-values before and after PSM between the two groups. Statistical analyses were carried out using Stata/SE version 16.0 (StataCorp., College Station, TX, USA). All presented *p*-values were two-sided, and *p*-values of <0.05 were considered statistically significant.

## 3. Results

### 3.1. Patient Characteristics

Table 1 shows the baseline characteristics of 211 patients who underwent RpLDG. The study subjects consisted of 143 patients in NOG and 68 in OG with a mean age of 65.2 years. All baseline variables were well-balanced, except those of the ASA physical status classification [16]. Patients in the OG had a significantly higher score than those in the NOG (ASA ≥ 2, 47.6% vs. 80.9%, *p* < 0.001). After PSM, this difference disappeared, and 68 matched pairs were completed.

### 3.2. Operative Results

Table 2 summarizes the operative details of the two groups. There were statistically significant differences in terms of operation time and estimated bleeding volume. The operation time in OG was longer than that in NOG (200.59 vs. 208.68 min, *p* = 0.044). The intraoperative bleeding volume was significantly higher in OG (40.96 vs. 58.01 mL, *p* = 0.001). Additionally, the number of harvested lymph nodes was lower in OG (36.81 vs. 32.13, *p* = 0.039), but the number of metastatic lymph nodes did not differ. There were no statistical differences in terms of surgical techniques and the degree of nodal dissection or omentectomy.

### 3.3. Hospital Course and Postoperative Complications

As shown in Table 3, there were no noticeable differences in hospital course and postoperative morbidity. In both groups, there was no difference in time to first flatus and diet resumption (*p* = 0.847, *p* = 0.992, respectively). The postoperative hospital stays were also similar (7.79 vs. 8.15, *p* = 0.824). There were no mortalities and no significant differences in rate and degree of morbidity (*p* = 1.000 in systemic complications, *p* = 1.000 in surgical complications).

Table 4 provides details of the postoperative complications. There were 21 complicated cases in NOG (14.7%) and 11 complicated cases in OG (16.2%), these are not statistically different (*p* = 0.778). Moreover, the rates of each complication did not differ significantly between the two groups. Univariate and multivariate analyses after PSM showed no significant risk factors for postoperative complications (Table 5).

Figure 1 shows the impact of overweight and obesity on the risk of postoperative complications for each subgroup. This analysis revealed that these conditions did not increase the risk of postoperative complications in any subgroup.

The incidence of postoperative complications did not differ significantly in the overweight and obese subgroup.

ASA = American Society of Anesthesiologists; EBV = estimated bleeding volume; OR = odds ratio; CI = confidence interval.

## 4. Discussion

Although several studies report that reduced-port laparoscopic gastrectomy is not inferior to conventional methods in postoperative or oncological outcomes, reduced-port surgery in obese patients has always been a challenge for gastric surgeons [7,8,17]. Despite our previous report that reduced port laparoscopic gastrectomy can be a safe alternative to conventional port gastrectomy in obese patients [18], we wanted to conduct a more supplementary study because we felt constant difficulties in reduced port gastrectomy in overweight and obese patients with gastric cancer.

In this study, 17 patients who underwent combined surgery and 42 patients who had ports added during surgery were excluded from the 270 patients who underwent RpLDG for gastric malignancy. Of the 17 patients who underwent combined surgery, nine had a BMI of less than 25 kg/m^2^, and eight had a BMI of more than 25 kg/m^2^. There were 15 cases of laparoscopic cholecystectomy, a single case of hepatic cyst, and another case of incidental gastric subepithelial tumor detected in the remnant stomach during RpLDG. These cases were excluded from the study because of combined surgery, which could affect outcomes such as operative time and bleeding volume. Of the 42 patients who needed an additional trocar, 27 were not overweight and 15 were overweight. In these cases, we added one assistant 5 mm port when adhesiolysis could not be completed with only two right-side working ports due to adhesion from the previous operation or to compress the protruded pancreas body and to better secure the operating field for D2 lymph node dissection. We aimed to investigate whether overweight or obesity had any significant effect on the short-term postoperative outcomes in patients who have successfully undergone RpLDG. Thus, we did not focus on the success rate of the procedure. Therefore, we excluded cases from the analysis in which a port was added during RpLDG.

In this study, PSM analysis was performed to minimize selection bias and balance the difference in baseline characteristics. As a result, operation time and estimated blood loss in the OG were significantly higher but did not affect intraoperative security or effectiveness. This illustrates the minimal impact of overweight and obesity that has been mentioned in many other studies [9,10]. Longer operation time and higher volume of blood loss in the OG showed similarly regardless of the type of surgical approaches, such as open, laparoscopy-assisted, totally laparoscopic, and reduced-port laparoscopic surgery [10,19,20,21]. Therefore, longer operation time and higher volume of blood loss in the OG do not need to be considered disadvantages of RpLDG.

The degree of nodal dissection between the two groups did not differ, but the number of harvested lymph nodes was statistically lower in the OG. This may be because the en-bloc basin resection is not easy in overweight and obese patients. A large amount of visceral fat is brittle and bleeds easily enough to obstruct the surgical field, which makes it difficult to dissect the lymph nodes while maintaining continuity. However, accurate N staging is more important than the number of harvested lymph nodes. According to the Japanese classification of gastric carcinoma, most gastric surgeons set the number of harvested lymph nodes required for N status determination to be 16 or more [22]. In this study, the mean number of harvested lymph nodes was 36.81 and 32.13, respectively, well above the recommended minimum. In addition, the number of lymph nodes with pathologically confirmed metastases did not show a significant intergroup difference. Therefore, the differences in the number of harvested lymph nodes seen in our study would not lead to a judgement that RpLDG in overweight and obese patients with gastric cancer was oncologically inferior compared to that in non-overweight patients.

Despite our promising results, we admit that reduced-port laparoscopic surgery in overweight and obese patients with gastric cancer is associated with technical challenges. However, the operative procedure of RpLDG is similar to that of conventional port laparoscopic gastrectomy, so it does not require special instruments or much surgical experience. Previously, we reported that approximately 30 cases are required for an experienced surgeon to acquire technical proficiency in reduced-port laparoscopic gastrectomy [4]. In comparison, Park et al. [23] reported that even a novice surgeon needed 20–30 operations to achieve technical proficiency with RpLDG, and that this requirement was not different from the learning curve of conventional port laparoscopic distal gastrectomy. Based on these results, we believe that the overweight or obesity in gastric cancer patients is not a decisive factor for the replacement of conventional methods for RpLDG. Above all, as mentioned by Lee et al. [8], the RpLDG has an advantage that it can be performed without an assistant, and this would be a way to cope with the lack of surgical assistance in many institutions in South Korea.

The present study has several limitations. First, the study was inherently limited by its retrospective and non-randomized design. Second, this study analyzed only short-term outcomes, precluding long-term outcomes. Finally, all operations were performed by a single experienced surgeon. This could have eliminated selection bias and the effect of the learning curve, but also limits the generalizability of the study results. Hence, the clinical benefits and oncologic safety of this procedure must be validated in a larger prospective randomized clinical trial with long term follow-up.

Authors should discuss the results and how they can be interpreted from the perspective of previous studies and of the working hypotheses. The findings and their implications should be discussed in the broadest context possible. Future research directions may also be highlighted.

## 5. Conclusions

In conclusion, the present study demonstrated that the short-term surgical outcomes of RpLDG in overweight and obese gastric cancer patients were comparable to those in non-overweight gastric cancer patients. Despite a longer operation time and higher blood loss in these patients, we believe that RpLDG is a safe and effective procedure since no significant differences were found in terms of operative results, postoperative outcomes, or morbidity and mortality. Our experience has shown that RpLDG can be safely performed in overweight and obese gastric cancer patients.

## Figures and Tables

**Figure 1 jcm-11-06453-f001:**
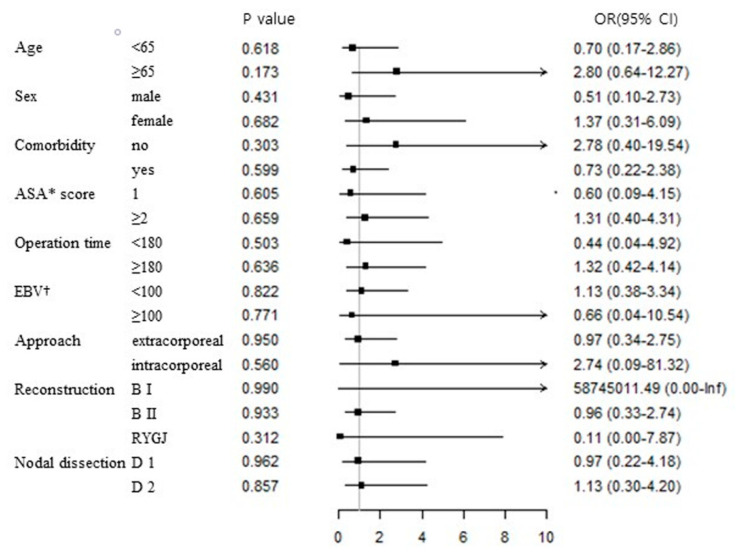
Subgroup analysis for impact of overweight and obesity on the risk of postoperative complications.

**Table 1 jcm-11-06453-t001:** Patient characteristics.

	Total Set (n = 211)			PSM Set (n = 136)		
	NOG (n = 143)	OG (n = 68)	*p-*Value	NOG (n = 68)	OG (n = 68)	*p*-Value
Body mass index (kg/m^2)^	22.07 ± 2.00	27.85 ± 2.75	<0.001	21.97 ± 2.11	27.85 ± 2.75	<0.001
Age (years)	66.18 ± 11.11	62.85 ± 11.03	0.043	63.21 ± 10.88	62.85 ± 11.03	0.851
Sex (male)	101 (70.6)	47 (69.1)	0.823	49 (72.1)	47 (69.1)	0.707
Abdominal surgery history	11 (7.7)	7 (10.3)	0.527	6 (8.8)	7 (10.3)	0.771
Comorbidity	78 (54.6)	50 (73.5)	0.008	52 (76.5)	50 (73.5)	0.692
Hypertension	51 (35.7)	38 (55.9)	0.005	40 (58.8)	38 (55.9)	0.729
Diabetes mellitus	28 (19.6)	16 (23.5)	0.509	15 (22.1)	16 (23.5)	0.838
Ischemic heart disease	6 (4.2)	5 (7.4)	0.335	4 (5.9)	5 (7.4)	0.73
Obstructive pulmonary disease	8 (5.6)	2 (2.9)	0.397	2 (2.9)	2 (2.9)	1
Cerebrovascular disease	4 (2.8)	0 (0.0)	0.164	2 (2.9)	0 (0.0)	0.154
Liver cirrhosis	2 (1.4)	1 (1.5)	0.967	1 (1.5)	1 (1.5)	1
Renal disease	1 (0.7)	1 (1.5)	0.589	1 (1.5)	1 (1.5)	1
ASA score			<0.001			0.568
1	75 (52.4)	13 (19.1)		17 (25.0)	13 (19.1)	
2	62 (43.4)	51 (75.0)		45 (66.2)	51 (75.0)	
3	5 (3.5)	4 (5.9)		5 (7.3)	4 (5.9)	
4	1 (0.7)	0 (0.0)		1 (1.5)	0 (0.0)	
Tumor size (mm)	22.10 ± 22.53	21.43 ± 9.17	0.15	20.54 ± 12.30	21.43 ± 9.17	0.282
Histological grade			0.793			0.72
Differentiated	92 (64.3)	45 (66.2)		43 (63.2)	45 (66.2)	
Undifferentiated	51 (35.7)	23 (33.8)		25 (36.8)	23 (33.8)	
Lymphovascular invasion	11 (7.7)	3 (4.4)	0.371	5 (7.4)	3 (4.4)	0.466
Perineural invasion	7 (4.9)	3 (4.4)	0.877	3 (4.4)	3 (4.4)	1
Depth of invasion			0.73			0.799
pTis	3 (2.1)	2 (2.9)		2 (2.9)	2 (2.9)	
pT1	130 (90.9)	64 (94.1)		63 (92.7)	64 (94.1)	
pT2	7 (4.9)	2 (2.9)		2 (2.9)	2 (2.9)	
pT3	1 (0.7)	0 (0.0)		1 (1.5)	0 (0.0)	
pT4	2 (1.4)	0 (0.0)		0 (0.0)	0 (0.0)	
Nodal metastasis			0.46			0.242
pN0	129 (90.2)	63 (92.7)		60 (88.2)	63 (92.7)	
pN1	11 (7.7)	2 (2.9)		7 (10.3)	2 (2.9)	
pN2	1 (0.7)	1 (1.5)		0 (0.0)	1 (1.5)	
pN3	2 (1.4)	2 (2.9)		1 (1.5)	2 (2.9)	

Values are presented as number (%) or mean ± standard deviation. PSM = propensity score matching; NOG = Non-overweight group; OG = Overweight group; ASA = American Society of Anesthesiologists.

**Table 2 jcm-11-06453-t002:** Operative details.

	Total Set (n = 211)			PSM Set (n = 136)		
	NOG (n = 143)	OG (n = 68)	*p*-Value	NOG (n = 68)	OG (n = 68)	*p*-Value
Operation time (min)	199.51 ± 39.71	208.68 ± 29.15	0.014	200.59 ± 40.39	208.68 ± 29.15	0.044
Estimated bleeding volume (mL)	42.59 ± 37.04	58.01 ± 45.82	<0.001	40.96 ± 35.40	58.01 ± 45.82	0.001
Approach			0.081			0.259
Intracorporeal	120 (83.9)	63 (92.7)		59 (86.8)	63 (92.7)	
Extracorporeal	23 (16.1)	5 (7.3)		9 (13.2)	5 (7.3)	
Reconstruction			0.06			0.13
Billroth I	18 (12.6)	2 (2.9)		8 (11.8)	2 (2.9)	
Billroth II	118 (82.5)	61 (89.7)		57 (83.8)	61 (89.7)	
Roux-en-Y	7 (4.9)	5 (7.4)		3 (4.4)	5 (7.4)	
Nodal dissection			0.172			0.167
D1+	74 (51.8)	42 (61.8)		34 (50.0)	42 (61.8)	
D2	69 (48.2)	26 (38.2)		34 (50.0)	26 (38.2)	
No. harvested lymph nodes	39.13 ± 15.88	32.13 ± 16.59	0.027	36.81 ± 16.45	32.13 ± 16.59	0.039
No. metastatic lymph nodes	0.40 ± 2.13	0.59 ± 2.99	0.607	0.41 ± 2.10	0.59 ± 2.99	0.425
Omentectomy			0.511			0.645
Complete	130 (90.9)	63 (92.7)		61 (89.7)	63 (92.7)	
Bursectomy	4 (2.8)	3 (4.4)		2 (2.9)	3 (4.4)	
Partial	9 (6.3)	2 (2.9)		5 (7.4)	2 (2.9)	
Proximal margin (mm)	53.03 ± 34.30	58.53 ± 89.44	0.662	57.28 ± 35.40	58.53 ± 89.44	0.205
Distal margin (mm)	54.98 ± 29.71	72.38 ± 63.05	0.158	54.81 ± 31.13	72.38 ± 63.05	0.176

Values are presented as number (%) or mean ± standard deviation. PSM = propensity score matching; NOG = Non-overweight group; OG = Overweight group.

**Table 3 jcm-11-06453-t003:** Postoperative outcomes.

	Total Set (n = 211)			PSM Set (n = 136)		
	NOG (n = 143)	OG (n = 68)	*p*-Value	NOG (n = 68)	OG (n = 68)	*p*-Value
Time to first flatus (d)	3.13 ± 0.95	3.07 ± 0.89	0.666	3.04 ± 0.89	3.07 ± 0.89	0.847
Time to diet resumption (d)	1.99 ± 0.19	2.06 ± 0.49	0.4	2.01 ± 0.12	2.06 ± 0.49	0.992
Length of hospital stay (d)	7.87 ± 3.60	8.15 ± 4.83	0.895	7.79 ± 3.80	8.15 ± 4.83	0.824
Fever	11 (7.7)	9 (13.2)	0.199	6 (8.8)	9 (13.2)	0.412
Transfusion	2 (1.4)	1 (1.5)	1	2 (2.9)	1 (1.5)	1
Severity of medical complication	3 (2.1)	2 (2.9)	0.658	2 (2.9)	2 (2.9)	1
Mild	1 (0.7)	0 (0.0)		1 (1.5)	0 (0.0)	
moderate	1 (0.7)	2 (2.9)		1 (1.5)	2 (2.9)	
Severe	1 (0.7)	0 (0.0)		0 (0.0)	0 (0.0)	
Severity of surgical complication	19 (13.3)	9 (13.2)	0.992	9 (13.2)	9 (13.2)	1
mild	16 (11.2)	5 (7.3)		7 (10.2)	5 (7.3)	
moderate	1 (0.7)	3 (4.4)		1 (1.5)	3 (4.4)	
severe	1 (0.7)	1 (1.5)		1 (1.5)	1 (1.5)	

Values are presented as number (%) or mean ± standard deviation. PSM = propensity score matching; NOG = Non-overweight group; OG = Overweight group.

**Table 4 jcm-11-06453-t004:** Postoperative complications.

	Total Set (n = 211)			PSM Set (n = 136)		
	NOG (n = 143)	OG (n = 68)	*p*-Value	NOG (n = 68)	OG (n = 68)	*p*-Value
Clavien-Dindo grade			0.117			0.592
I	17 (11.9)	5 (7.3)		8 (11.8)	5 (7.3)	
II	2 (1.4)	5 (7.3)		2 (2.9)	5 (7.3)	
III	2 (1.4)	1 (1.5)		1 (1.5)	1 (1.5)	
IV	0 (0.0)	0 (0.0)		0 (0.0)	0 (0.0)	
Medical complications			0.100			0.333
Cardiovascular	0 (0.0)	1 (1.5)		0 (0.0)	1 (1.5)	
Pulmonary	3 (2.1)	0 (0.0)		2 (2.9)	0 (0.0)	
Renal	0 (0.0)	1 (1.5)		0 (0.0)	1 (1.5)	
Surgical complications			0.822			0.772
Ascites (Lymphorrhea)	1 (0.7)	2 (2.9)		1 (1.5)	2 (2.9)	
Chyle leakage	1 (0.7)	0 (0.0)		0 (0.0)	0 (0.0)	
Gastric stasis	5 (3.5)	1 (1.5)		1 (1.5)	1 (1.5)	
Ileus	6 (4.2)	4 (5.9)		5 (7.3)	4 (5.9)	
Intraabdominal infection	3 (2.1)	2 (2.9)		0 (0.0)	2 (2.9)	
Pancreatitis	1 (0.7)	0 (0.0)		1 (1.5)	0 (0.0)	
Wound infection	1 (0.7)	0 (0.0)		1 (1.5)	0 (0.0)	
Total	21 (14.7)	11 (16.2)	0.778	11 (16.2)	11 (16.2)	1

Values are presented as number (%). PSM = propensity score matching; NOG = Non-overweight group; OG = Overweight group.

**Table 5 jcm-11-06453-t005:** Univariate and multivariate analysis of risk factors for postop complications.

	PSM Set (n = 136)			
	Univariate		Multivariate	
	OR (95% CI)	*p*-Value	OR (95% CI)	*p*-Value
Age (≥65 years)	0.96 (0.38–2.40)	0.929	1.35 (0.49–3.74)	0.563
Sex (female)	0.88 (0.32–2.45)	0.81	0.85 (0.29–2.53)	0.775
Body mass index (≥25 kg/m^2^)	1.00 (0.40–2.49)	1	1.05 (0.39–2.87)	0.92
Comorbidity	0.67 (0.25–1.80)	0.422	1.10 (0.28–4.32)	0.89
ASA score (≥2)	0.54 (0.20–1.48)	0.233	0.55 (0.14–2.07)	0.374
Operation time (≥3 h)	0.86 (0.29–2.57)	0.787	0.73 (0.22–2.38)	0.6
Estimated bleeding volume (≥100 cc)	1.13 (0.30–4.31)	0.86	1.20 (0.27–5.28)	0.805
Approach				
Intracorporeal	1		1	
Extracorporeal	1.48 (0.38–5.80)	0.575	4.03 (0.37–43.65)	0.251
Reconstruction				
Billroth I	1		1	
Billroth II	1.73 (0.21–14.44)	0.614	4.76 (0.19–119.10)	0.342
Roux-en-Y	3.00 (0.22–40.93)	0.41	6.94 (0.29–167.53)	0.233
Nodal dissection				
D1+	1		1	
D2	2.06 (0.81–5.21)	0.127	2.01 (0.74–5.49)	0.171

PSM = propensity score matching; NOG = Non-overweight group; OG = Overweight group; ASA = American Society of Anesthesiologists; OR = odds ratio; CI = confidence interval.

## Data Availability

Data are available from the corresponding author on reasonable request.

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
