# Peer review of "The Impact of Overweight and Obesity on Reduced-Port Laparoscopic Distal Gastrectomy for Gastric Cancer Patients: A Propensity Score Matching Analysis of a Single-Institution Data"

_jcm, 2022, doi:10.3390/jcm11216453_

Round 1

Reviewer 1 Report

Thank you for the opportunity to review this intriguing work on short-term postoperative outcomes of reduced-port laparoscopic distal gastrectomy in overweight and obese patients with gastric cancer. This is a well-done study with sound methodology and moderate to high clinical significance for readers of JCM. With minor revisions:

In the abstract, please give exact p-values instead of p<0.05. At the end of the sentences starting with "In the subgroup analysis..." and "We demonstrated comparable short-term..." in the abstract, please provide a p-value and/or brief percentage comparison of the most salient outcomes in parentheses. 

I have no major recommendations for revision or concerns for how the study was performed or interpreted.

Author Response

Point 1: Thank you for the opportunity to review this intriguing work on short-term postoperative outcomes of reduced-port laparoscopic distal gastrectomy in overweight and obese patients with gastric cancer. This is a well-done study with sound methodology and moderate to high clinical significance for readers of JCM. With minor revisions:

In the abstract, please give exact p-values instead of p<0.05. At the end of the sentences starting with "In the subgroup analysis..." and "We demonstrated comparable short-term..." in the abstract, please provide a p-value and/or brief percentage comparison of the most salient outcomes in parentheses. 

Response 1: Thanks for the in-deep review of the article. As you pointed out, we provided a p-value and/or brief percentage at the end of each sentence in parentheses.

I have no major recommendations for revision or concerns for how the study was performed or interpreted.

Point 2: English language and style- Moderate English changes required

Response 2: Thanks for the reviewer's comments.
An editing certificate is attached. Please see the attachment.

Reviewer 2 Report

The authors of this study try to present the evidence regarding the use of reduced-port laparoscopic gastrectomy in overweight/obese patients through a propensity-matched analysis. Although the topic of this manuscript is not entirely novel, but I've found the topic is interesting enough and can add more evidence into the existing literatures. The Introduction section described by the authors have addressed enough rationale or background to justify this study. The Methods section is robust and has been described in details to allow for replication of study. The presentation of the Results and Discussion is also good enough.

Author Response

Point 1: The authors of this study try to present the evidence regarding the use of reduced-port laparoscopic gastrectomy in overweight/obese patients through a propensity-matched analysis. Although the topic of this manuscript is not entirely novel, but I've found the topic is interesting enough and can add more evidence into the existing literatures. The Introduction section described by the authors have addressed enough rationale or background to justify this study. The Methods section is robust and has been described in details to allow for replication of study. The presentation of the Results and Discussion is also good enough.

Response 1: I deeply appreciate your in-depth revision and good evaluation.

Reviewer 3 Report

In this paper the Authors aim to assess the short-term postoperative outcomes of reduced-port laparoscopic distal gastrectomy and demonstrate its safety and feasibility in overweight and obese patients with gastric cancer.. It is an interesting and debated topic. A comprehensive and extensive literature review of the NCBI database PubMed was also carried out. The article was well conducted and it is interesting in its fields. It is a well-structured paper, written in good English and the References are up dated. 

Minor issues:

In the “discussion” section I suggest to better analyze the complications related to the obesity. Therefore, the following paper should be considered:

“Del Genio G, Tolone S, Gambardella C, Brusciano L, Volpe ML, Gualtieri G, Del Genio F, Docimo L. Sleeve Gastrectomy and Anterior Fundoplication (D-SLEEVE) Prevents Gastroesophageal Reflux in Symptomatic GERD. Obes Surg. 2020 May;30(5):1642-1652. doi: 10.1007/s11695-020-04427-1. PMID: 32146568.”

“Pizza F, Lucido FS, D'Antonio D, Tolone S, Gambardella C, Dell'Isola C, Docimo L, Marvaso A. Biliopancreatic Limb Length in One Anastomosis Gastric Bypass: Which Is the Best? Obes Surg. 2020 Oct;30(10):3685-3694. doi: 10.1007/s11695-020-04687-x. PMID: 32458362.” 

Author Response

Point 1: In this paper the Authors aim to assess the short-term postoperative outcomes of reduced-port laparoscopic distal gastrectomy and demonstrate its safety and feasibility in overweight and obese patients with gastric cancer. It is an interesting and debated topic. A comprehensive and extensive literature review of the NCBI database PubMed was also carried out. The article was well conducted and it is interesting in its fields. It is a well-structured paper, written in good English and the References are up dated. 

Response 1: Thanks for the in-deep review of this paper.

Point 2: Minor issues:

In the “discussion” section I suggest to better analyze the complications related to the obesity. Therefore, the following paper should be considered:

Response 2: I agree with you. In this study, further analysis of obesity-related complications is expected to improve the quality of the paper. I read the two articles you presented carefully. The two articles were excellent in understanding the difference in outcomes according to bariatric surgical procedures.

Referring to the article you recommended, we investigated the frequency of the complication of this study according to the Clavien-Dindo classification grade and obtained the p-value, and changed table 4. Thank you for your in-depth review.